# Hysterectomies are associated with an increased risk of osteoporosis and bone fracture: A population-based cohort study

Ying-Ting Yeh[1], Pei-Chen Li[2], Kun-Chi Wu[3], Yu-Cih Yang[4,5], Weishan Chen[4,5], Hei-Tung Yip[4,5], Jen-Hung Wang[6], Shinn-Zong Lin[7], Dah-Ching Ding[2,8]*

1 Department of Physical Medicine and Rehabilitation Medicine, Hualien Tzu Chi Hospital, Buddhist Tzu Chi Medical Foundation, and Tzu Chi University, Hualien, Taiwan, 2 Department of Obstetrics and Gynecology, Hualien Tzu Chi Hospital, Buddhist Tzu Chi Medical Foundation, and Tzu Chi University, Hualien, Taiwan, 3 Department of Orthopedics, Hualien Tzu Chi Hospital, Buddhist Tzu Chi Medical Foundation, and Tzu Chi University, Hualien, Taiwan, 4 Management Office for Health Data, China Medical University Hospital, Taichung, Taiwan, 5 College of Medicine, China Medical University, Taichung, Taiwan, 6 Department of Research, Hualien Tzu Chi Hospital, Buddhist Tzu Chi Medical Foundation, and Tzu Chi University, Hualien, Taiwan, 7 Department of Neurosurgery, Hualien Tzu Chi Hospital, Buddhist Tzu Chi Medical Foundation, and Tzu Chi University, Hualien, Taiwan, 8 Institute of Medical Sciences, Tzu Chi University, Hualien, Taiwan

* dah1003@yahoo.com.tw

**Data Availability Statement:** All relevant data were showed in the manuscript.

**Funding:** This study is supported in part by the Taiwan Ministry of Health and Welfare Clinical Trial

## Abstract

### Aim

This study investigated the risk of osteoporosis or bone fractures (vertebrae, hip and others) in hysterectomized women in Taiwan.

### Materials and methods

This is a retrospective population-based cohort study from 2000 to 2013. Women aged ≥30 years who underwent hysterectomy between 2000 and 2012 were included in this study. The comparison group was randomly selected from the database with a 1:4 matching with age and index year. Incidence rate and hazard ratios of osteoporosis and bone fracture between hysterectomized women and the comparison group were calculated. Cox proportional hazard regressions were used to calculate hazard ratios (HRs) and 95% confidence intervals (CIs).

### Results

We identified 9,189 hysterectomized women and 33,942 age-matched women without a hysterectomy. All women were followed for a median time of about 7 years. The adjusted hazard ratio (aHR) of subsequent osteoporosis or bone fracture was higher in the hysterectomy women (2.26, 95% confidence interval [CI] = 2.09–2.44) than in the comparison group. In the subgroup analysis, oophorectomy and estrogen therapy increase the risk of osteoporosis or fracture in both groups. Regarding the fracture site, the aHR of vertebral fracture (4.92, 95% CI = 3.78–6.40) was higher in the hysterectomized women than in the comparison group. As follow-up time increasing, the aHR of vertebral fracture in hysterectomized

Center (MOHW108-TDU-B-212- 133004), China Medical University Hospital, Academia Sinica Stroke Biosignature Project (BM10701010021), MOST Clinical Trial Consortium for Stroke (MOST 107-2321-B-039 -004-), Tseng-Lien Lin Foundation, Taichung, Taiwan, and Katsuzo and Kiyo Aoshima Memorial Funds, Japan.

**Competing interests:** The authors have declared that no competing interests exist.

**Abbreviations:** CI, confidence interval; HR, hazard ratio; LHID, Longitudinal Health Insurance Database; MrOS, Osteoporotic Fractures in Men; TNHI, Taiwan National Health Insurance; NHIRD, National Health Insurance Research Database; OR, odds ratio; CCI, Charlson comorbidity index; BSO, bilateral salpingo-oophorectomy; SIR, standardized incidence ratio; ET, estrogen therapy; HT, Hormone therapy; AMH, anti-Mullerian hormone; IRR, incidence rate ratio.

women were 4.33 (95% CI = 2.99–6.28), 3.89 (95% CI = 2.60–5.82) and 5.42 (95% CI = 2.66–11.01) for <5, 5–9 and ≥9 years of follow-up, respectively.

## Conclusions

In conclusion, we found that hysterectomized women might be associated with increased risks of developing osteoporosis or bone fracture.

## Introduction

Osteoporosis and its associated fragility fractures are a significant global issue with an impact on humans second only to cardiovascular disease [1,2]. Osteoporosis is a skeletal system disease that reduces bony mass and disrupts the bone structure, causing decreased bone strength and leading to fragility fractures. Moreover, women were found to have double the risk for osteoporosis and triple the risk for fragility fractures compared with men at age 50 [3]. Furthermore, fractures are notorious for increased mortality, morbidity, disabilities in daily living, social costs, and psychogenic problems [4].

Hysterectomy, a surgery to remove the uterus, is the most common gynecologic operation worldwide, including in the United States and Taiwan [5–7]. Hysterectomy is thought to be related to multiple comorbidities because it might be related to earlier physiological menopause than in the general population, which results in earlier hormonal changes and may be related to osteoporosis and bone fractures [8].

Since osteoporosis and bone fracture have a strong relationship with menopause and hormone changes, we hypothesized that hysterectomy may increase the risk of osteoporosis and bone fracture. However, there are scarce studies discussing the association between hysterectomy and osteoporosis or fracture. The previous study showed hysterectomy was associated with bone loss, however, the study sample size was small [9]. Another study also with a small sample size showed hysterectomy associated with a decreased bone mineral density in the lumbar spine and hip [10]. There were two population-based studies regarding the relationship between hysterectomy and long-term osteoporotic fracture or bone mineral density [11,12]. However, there was no study discussing both outcomes together.

This retrospective study used the Taiwan National Health Insurance (TNHI) Database of one million randomly sampling cohort from a total of 23 million people in Taiwan to investigate the risk of developing osteoporosis and bone fracture for women with hysterectomy.

## Materials and methods

### Data source

This retrospective cohort study was conducted using claims data from the Longitudinal Health Insurance Database 2000 (LHID 2000), which is a subset of the National Health Insurance Research Database (NHIRD). The NHIRD was built by the National Health Research Institute; it contains 23 million NHI enrollees, which includes approximately 99% of the population of Taiwan. More than 20,000 medical care facilities, including hospitals, clinics, and pharmacies, which represent over 93% of all healthcare facilities in Taiwan, were contracted by the NHI project. The NHIRD includes outpatient and inpatient information about medication use, surgical procedures, intervention procedures, and clinical prescriptions. The NHRI claims that there are no statistically significant differences in the data on age, geographic region, and

health care costs from the LHID 2000 and all claims data. Disease diagnoses were identified by the International Classification of Diseases, 9th Revision, Clinical Modification (ICD-9-CM). This database has been validated by many studies [13–15] and proved the correct coding and disease. This study was approved by the Institutional Review Board of China Medical University and the Hospital Research Ethics Committee (IRB permit number: CMUH-104-REC2-115) and is in compliance with institutional guidelines. The written informed consent was waived due to low risk and approved by the institutional IRB.

## Sample participants

We retrospectively examined the hysterectomy and matched non-hysterectomy cohorts to investigate the relationship between hysterectomy and the risk of osteoporosis or bone fracture (ICD-9-CM code 800–829). The hysterectomy cohort included women aged ≥30 years who underwent hysterectomy (NHI claim codes 97020K, 97021A, 97022B, 97025K, 97026A, 97027B, 97027C, 97035K, 97036A, and 97037B) between January 1, 2000 and December 31, 2012. We defined the first date undergoing a hysterectomy in the study period as the index year. The comparison cohort was randomly selected from those beneficiaries without hysterectomy with matching by age (±5 years), and index year at a ratio of 1:4 with frequency matching. Women under 30 years old or above 100 years old and with a history of osteoporosis, bone fracture and oophorectomy before the index date were excluded from the present study. Patients with bone fracture that caused by vehicle injury and falls were also eliminated. Both cohorts were followed-up until the women developing osteoporosis, fracture, death, withdrew from the NHI program, or December 31, 2013, whichever occurred first. This study also considered confounding factors, such as the urbanization of residence, monthly income, occupation, and Charlson comorbidity index (CCI) [16]. We categorized into 4 levels of urbanization of where a subject lived (level 1, most urbanized; level 4, least urbanized). How the income and urbanization affect the medical resources used have been reported [17,18]. The study flow chart is illustrated in Fig 1.

## Outcomes

The outcomes in this study were osteoporosis (NHI claim codes 733.0) or fracture (NHI claim codes 733.1, 800–804, 807–819, 822–829); vertebral fracture (NHI claim codes 805–806); hip fracture (NHI claim codes 820–821) diagnosed with 2 times of clinic visits and one time of hospitalization. In Taiwan, the diagnosis of osteoporosis was made by a Dual-energy X-ray absorptiometry (DEXA) exam. The other diagnostic modalities are history (age, menopause) and plain X-ray of vertebrae, hip or wrist.

## Comorbidities

We also considered whether the women had an increased risk of osteoporosis or fracture due to undergoing unilateral (NHI claim codes 80802C, 80807C) or bilateral (NHI claim codes 80807B, 80811C, 80812C, 80602B, 80602C, and 80802B) oophorectomy. Associated comorbidities were also considered potential confounding factors to determine associations between women with or without hysterectomy. The CCI was used to determine the severity of comorbidities in this study. The CCI score is a widely used clinical index for a variety of disorders and cancers [19,20]. The higher the CCI, the more severe the comorbidities. We also included prescriptions for Estradiol and Premarin in the database (ATC codes G03C) during the study period. Women were considered as estrogen therapy (ET) users if they received in-hospital estrogen therapy for more than 30 days.

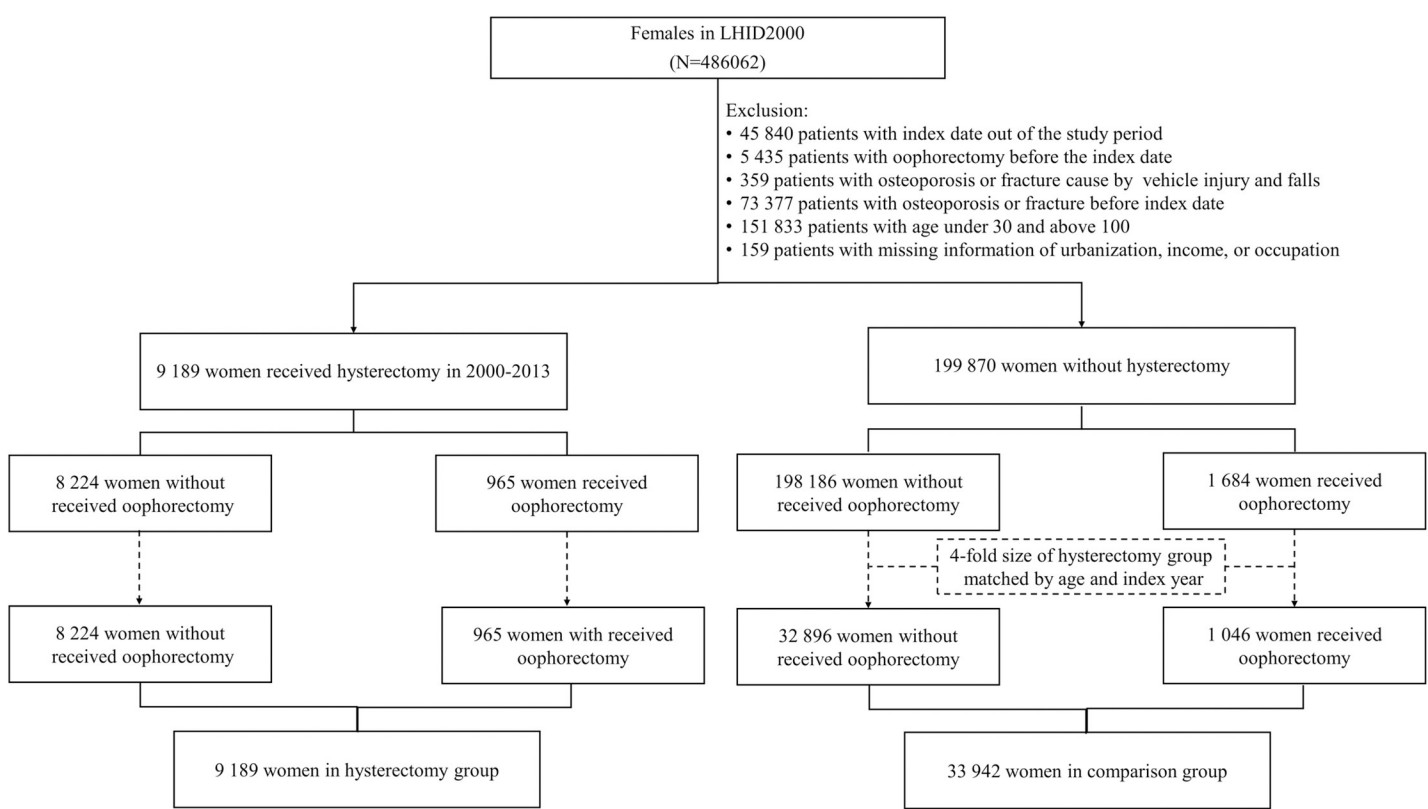

**Fig 1. Study flow chart: The participants population enrolled from the National Health Insurance Research Database.**

## Statistical analysis

We used standard mean difference (SMD), which indicates there was a neglected difference when SMD <0.1, to examine the differences between categorical and continuous baseline characteristics, such as age, gender, urbanization of residence, monthly income, occupation, and CCI. Cox proportional hazard regressions were used to calculate hazard ratios (HRs) and 95% confidence intervals (CIs). To evaluate the proportional hazard assumption of the Cox regression model, we added an interaction term between the study groups (hysterectomy/comparison group) and the logarithm of age in the Cox regression model. We also performed an analysis stratified by follow-up duration. The cumulative incidence of osteoporosis or bone fracture survival curves were plotted by the Kaplan–Meier estimator, and the log-rank test was used to evaluate the difference between the two groups. We used univariable and multivariable Poisson regression analysis to estimate the incidence rate ratio (IRR) and 95% CI of two groups. All statistical analyses were performed using SAS version 9.4 software (SAS Institute, Cary, NC, USA). The significance threshold was set at 0.05 for a two-tailed p-value.

## Results

### Subject characteristics

Fig 1 shows the flowchart used to select the hysterectomy and comparison groups from the NHIRD. After frequency matching, our study cohort consisted of 43,131 women. The hysterectomy cohort included 9,189 women and the comparison cohort included 33,711 women (Fig 1). The study subjects were predominantly insured persons from 40 to 49 years old (61%)

**Table 1. Baseline characteristics in women with and without hysterectomy.**

| | Hysterectomy | Comparison group | |
|---|---|---|---|
| | (n = 9 189) | (n = 33 942) | SMD |
| Age, years | | | |
| 30–39 | 1285 (13.98) | 5063 (14.92) | 0.03 |
| 40–49 | 5692 (61.94) | 20795 (61.27) | 0.01 |
| 50–59 | 1481 (16.12) | 5348 (15.76) | 0.01 |
| ≧60 | 731 (7.96) | 2736 (8.06) | 0.004 |
| median (Q1-Q3) | 45.45 (42.22–49.78) | 44.24 (41.42–49.77) | 0.03 |
| Urbanization | | | |
| 1 | 3066 (33.37) | 11972 (35.27) | 0.04 |
| 2 | 2797 (30.44) | 10158 (29.93) | 0.01 |
| 3 | 1518 (16.52) | 5532 (16.3) | 0.01 |
| 4 | 1808 (19.68) | 6280 (18.5) | 0.03 |
| Insurance premium | | | |
| 0~15000 | 2067 (22.49) | 8293 (24.43) | 0.05 |
| 15000~30000 | 5434 (59.14) | 19181 (56.51) | 0.05 |
| ≧30000 | 1688 (18.37) | 6468 (19.06) | 0.02 |
| Occupation | | | |
| White collar | 4853 (52.81) | 19442 (57.28) | 0.08 |
| Blue collar | 3715 (40.43) | 12146 (35.78) | 0.09 |
| Other | 621 (6.76) | 2354 (6.94) | 0.09 |
| Oophorectomy | 965 (10.50) | 1046 (3.08) | 0.30 |
| Charlson comorbidity index | | | |
| 0 | 8323 (90.58) | 31913 (94.02) | 0.13 |
| 1 | 355 (3.86) | 1039 (3.06) | 0.04 |
| ≧2 | 511 (5.56) | 990 (2.92) | 0.13 |
| Estrogen treatment | | | |
| Estradiol and premarin | 38 (0.41) | 59 (0.17) | 0.04 |
| Follow up times, year | | | |
| median (Q1-Q3) | 6.66 (3.78–10.14) | 7.32 (4.26–10.67) | 0.12 |

SMD: Standard mean difference.

who lived in a high degree of urbanization, had an insurance premium between 15,000 and 30,000, and had a white-collar occupation. A total of 965 and 1,046 women underwent oophorectomy in the hysterectomy cohort and comparison cohort, respectively. The hysterectomy cohort had a higher proportion of women with a CCI score of 2 than the comparison cohort (Table 1). The median follow-up time was 6.66 and 7.32 year in the hysterectomy group and the comparison group, respectively.

## Risk of osteoporosis or fracture

In the hysterectomy group (n = 1049), the years (mean [SD]) from the index date through outcome was 4.97 [3.01] years, while in the comparison group (n = 1867), there were 5.09 [3.02] years.

Table 2 shows the incidence and HR of osteoporosis or fracture in women with hysterectomy compared to those without hysterectomy. Overall, the risk of osteoporosis or bone fracture in women with hysterectomy was significantly higher than that in the comparison group

**Table 2. Incidence rate and hazard ratio of osteoporosis or bone fracture in women with hysterectomy and oophorectomy compared with the comparison group.**

| | | N | Event | PY | IR | Crude HR (95% CI) | p-value | Adjusted* HR (95% CI) | p-value |
|---|---|---|---|---|---|---|---|---|---|
| **Osteoporosis or bone fracture** | | | | | | | | | |
| Hysterectomy | Oophorectomy | | | | | | | | |
| No | No | 32896 | 1776 | 243262 | 7.30 | 1.00 (reference) | | 1.00 (reference) | |
| No | unilateral | 477 | 36 | 3503 | 10.28 | **1.40** (1.01,1.95) | 0.04 | **1.63** (1.17,2.27) | 0.004 |
| No | bilateral | 569 | 55 | 5128 | 10.73 | **1.44** (1.10,1.89) | 0.01 | **2.04** (1.55,2.67) | <0.001 |
| Yes | No | 8224 | 954 | 57885 | 16.48 | **2.27** (2.10,2.45) | <0.001 | **2.26** (2.09,2.44) | <0.001 |
| Yes | unilateral | 531 | 22 | 2608 | 8.44 | 1.21 (0.80,1.85) | 0.37 | 1.30 (0.86,1.99) | 0.22 |
| Yes | bilateral | 434 | 73 | 3727 | 19.59 | **2.66** (2.10,3.36) | <0.001 | **2.95** (2.33,3.73) | <0.001 |
| **Osteoporosis** | | | | | | | | | |
| Hysterectomy | Oophorectomy | | | | | | | | |
| No | No | 32242 | 1122 | 243262 | 4.61 | 1.00 (reference) | | 1.00 (reference) | |
| No | unilateral | 453 | 12 | 3503 | 3.43 | 0.77 (0.43,1.35) | 0.36 | 0.94 (0.53,1.65) | 0.82 |
| No | bilateral | 534 | 20 | 5128 | 3.90 | 0.85 (0.55,1.32) | 0.47 | 1.36 (0.87,2.12) | 0.17 |
| Yes | No | 7655 | 385 | 57885 | 6.65 | **1.50** (1.34,1.69) | <0.001 | **1.52** (1.36,1.71) | <0.001 |
| Yes | unilateral | 517 | 8 | 2608 | 3.07 | 0.70 (0.35,1.40) | 0.31 | 0.76 (0.38,1.52) | 0.44 |
| Yes | bilateral | 389 | 28 | 3727 | 7.51 | **1.72** (1.18,2.50) | 0.005 | **1.96** (1.34,2.85) | <0.001 |
| **Hip fracture** | | | | | | | | | |
| Hysterectomy | Oophorectomy | | | | | | | | |
| No | No | 31198 | 78 | 243262 | 0.32 | 1.00 (reference) | | 1.00 (reference) | |
| No | unilateral | 442 | 1 | 3503 | 0.29 | 0.90 (0.13,6.47) | 0.92 | 1.08 (0.15,7.80) | 0.94 |
| No | bilateral | 517 | 3 | 5128 | 0.59 | 1.86 (0.59,5.91) | 0.29 | **3.27** (1.02,10.51) | 0.05 |
| Yes | No | 7288 | 18 | 57885 | 0.31 | 1.02 (0.61,1.70) | 0.95 | 1.07 (0.64,1.79) | 0.80 |
| Yes | unilateral | 509 | 2 | 2608 | 0.77 | – | | – | |
| Yes | bilateral | 363 | 2 | 3727 | 0.54 | 1.88 (0.46,7.64) | 0.38 | 2.75 (0.67,11.28) | 0.16 |
| **Vertebral fracture** | | | | | | | | | |
| Hysterectomy | Oophorectomy | | | | | | | | |
| No | No | 31225 | 105 | 243262 | 0.43 | 1.00 (reference) | | 1.00 (reference) | |
| No | unilateral | 447 | 6 | 3503 | 1.71 | **4.03** (1.77,9.17) | <0.001 | **4.64** (2.03,10.6) | <0.001 |
| No | bilateral | 517 | 3 | 5128 | 0.59 | 1.33 (0.42,4.20) | 0.62 | 1.79 (0.56,5.68) | 0.32 |
| Yes | No | 7390 | 120 | 57885 | 2.07 | **5.02** (3.86,6.52) | <0.001 | **4.92** (3.78,6.40) | <0.001 |
| Yes | unilateral | 510 | 1 | 2608 | 0.38 | 0.98 (0.14,7.00) | 0.98 | 1.05 (0.15,7.56) | 0.96 |
| Yes | bilateral | 367 | 6 | 3727 | 1.61 | **3.87** (1.70,8.80) | 0.001 | **4.50** (1.97,10.29) | <0.001 |
| **Other bone fracture** | | | | | | | | | |
| Hysterectomy | Oophorectomy | | | | | | | | |
| No | No | 31599 | 479 | 243262 | 1.97 | 1.00 (reference) | | 1.00 (reference) | |
| No | unilateral | 459 | 18 | 3503 | 5.14 | **2.59** (1.62,4.14) | <0.001 | **2.94** (1.83,4.71) | <0.001 |
| No | bilateral | 544 | 30 | 5128 | 5.85 | **2.92** (2.02,4.22) | <0.001 | **3.59** (2.47,5.21) | <0.001 |
| Yes | No | 7723 | 453 | 57885 | 7.83 | **4.04** (3.56,4.60) | <0.001 | **3.97** (3.49,4.51) | <0.001 |
| Yes | unilateral | 522 | 13 | 2608 | 4.98 | **2.60** (1.50,4.52) | <0.001 | **2.66** (1.53,4.61) | <0.001 |
| Yes | bilateral | 398 | 37 | 3727 | 9.93 | **5.17** (3.70,7.22) | <0.001 | **5.44** (3.89,7.61) | <0.001 |

PY: Person-years; IR: Incidence rate per 1,000 person-years; HR: Hazard ratio; CI: Confidence interval.

*: Model was adjusted for age, urbanization, insurance premium, occupation, estrogen treatment, and Charlson comorbidity index.

[incidence rate (IR) = 7.3 per 1,000 person-years vs 16.4 per 1,000 person-years; adjusted HR (aHR) = 2.26, 95% CI = 2.09–2.44].

For osteoporosis or bone fracture, compared with the women without hysterectomy and oophorectomy, the women with only hysterectomy had a higher risk of 2.26-fold (95% CI = 2.09–2.44). The risk of osteoporosis in women with hysterectomy only was also significantly higher than that in the comparison group (aHR = 1.52, 95% CI = 1.36–1.71).

For hip fracture, the hysterectomy had no higher risk than the comparison cohort (aHR = 1.07, 95% CI = 0.64–1.79). For vertebral fracture, the hysterectomy only had a higher risk than the comparison cohort (aHR = 4.92, 95% CI = 3.78–6.40). For other bone fracture, the risk in women with hysterectomy only was 3.97-fold (95%CI = 3.49–4.51) higher than that of the comparison cohort.

Bilateral oophorectomies also associated with an increased risk of osteoporosis and bone fracture in the comparison group and in the hysterectomy group (aHR = 2.04, 95% CI = 1.55–2.67; aHR = 2.95, 95% CI = 2.33–3.73, respectively).

## Subgroup analysis of the risk of osteoporosis or fracture with age

The incidence rate and HR of osteoporosis or fracture in women with hysterectomy or oophorectomy stratified by age are shown in Table 3.

For osteoporosis, the hysterectomy only women had a higher risk than the women without hysterectomy and oophorectomy in age group 30 to 39 (aHR = 2.31, 95% CI = 1.39–3.84), age group 40 to 49 (aHR = 2.03, 95% CI = 1.38–2.99) and in those over 60 years old (aHR = 1.37, 95% CI = 1.08–1.74). The women with hysterectomy plus oophorectomy also had a higher risk than the women without hysterectomy and oophorectomy in 40- to 49-year-olds (aHR = 2.03, 95% CI = 1.38–2.99).

For hip, the hysterectomy only women a higher risk than the women without hysterectomy and oophorectomy in age group 30–39 (aHR = 7.87, 95% CI = 1.20, 51.62).

The risk of vertebral fracture and other bone fracture were higher in women with hysterectomy only than those without hysterectomy among all age groups.

## Risk of vertebral and other bone fractures

Tables 4 and 5 present the risks of vertebral fracture and other fractures in women with hysterectomy compared with the comparison group stratified by follow-up year.

For the vertebral fracture, compared with the comparison cohort, the hysterectomy cohort had a 4.33-fold (95% CI = 2.99–6.28) higher risk in five years or shorter of follow up time, 3.89-fold (95% CI = 2.60–5.82) after five to nine years of follow-up and a 5.42-fold (95% CI = 2.66–11.01) higher risk after more than nine years of follow-up (Table 4).

For other fractures, the adjusted hazards were about 3.6 to 3.7 in every follow-up periods. (Table 5).

Fig 2 presents the cumulative incidence of each event shown by the Kaplan–Meier curves for osteoporosis, hip fracture, vertebral fracture, and other fractures outcomes.

## Stratified analysis between ET and hysterectomy on the risk of osteoporosis and fracture

Table 6 attempted to evaluate the interaction of ET and hysterectomy on the risk of osteoporosis and fracture. Women with ET alone increased risk of osteoporosis or bone fracture (adjusted incidence rate ratio [IRR] = 3.49, 95% CI = 2.10–5.81). Moreover, the adjusted IRR of hip fracture was increased to 9.59 (95% CI = 2.33–39.58) and that of vertebral fracture was increase to 26.33 (95% CI = 11.44, 60.60) in women with ET only. For women with both hysterectomy and ET, the adjusted IRR of Osteoporosis or bone fracture and Other bone fracture were 2.74 (95% CI = 1.23–6.12) and 6.73 (95% CI = 2.51–18.05), respectively.

**Table 3. Adjusted hazard ratio and 95% confidence interval of osteoporosis or fracture between women with hysterectomy or oophorectomy stratified by age.**

| | Comparison group without oophorectomy | Comparison group with oophorectomy | | Hysterectomy only | | Hysterectomy with oophorectomy | |
|---|---|---|---|---|---|---|---|
| | HR (95% CI) | HR* (95% CI) | p-value | HR* (95% CI) | p-value | HR* (95% CI) | p-value |
| *Osteoporosis* | | | | | | | |
| All | 1.00 (reference) | 0.84 (0.59,1.2) | 0.35 | **1.44** (1.28,1.61) | <0.001 | 1.30 (0.94,1.82) | 0.12 |
| Age | | | | | | | |
| 30–39 | 1.00 (reference) | 0.77 (0.24,2.48) | 0.66 | **2.31** (1.39,3.84) | 0.001 | 1.00 (0.14,7.28) | 1.00 |
| 40–49 | 1.00 (reference) | 1.05 (0.61,1.82) | 0.86 | **1.67** (1.42,1.97) | <0.001 | **2.03** (1.38,2.99) | <0.001 |
| 50–59 | 1.00 (reference) | **2.46** (1.43,4.22) | 0.001 | 1.18 (0.90,1.56) | 0.23 | 0.84 (0.37,1.88) | 0.67 |
| ≧60 | 1.00 (reference) | 0.41 (0.1,1.65) | 0.21 | **1.37** (1.08,1.74) | 0.01 | 0.55 (0.14,2.23) | 0.41 |
| *Hip fracture* | | | | | | | |
| All | 1.00 (reference) | 1.52 (0.56,4.17) | 0.41 | 0.92 (0.55,1.54) | 0.74 | 1.00 (0.25,4.08) | 1.00 |
| Age | | | | | | | |
| 30–39 | 1.00 (reference) | – | | **7.87** (1.20,51.62) | 0.03 | – | |
| 40–49 | 1.00 (reference) | – | | 1.19 (0.48,2.95) | 0.70 | 1.90 (0.26,14.14) | 0.53 |
| 50–59 | 1.00 (reference) | 4.50 (0.56,36.26) | 0.16 | 0.91 (0.20,4.15) | 0.90 | 3.09 (0.38,24.87) | 0.29 |
| ≧60 | 1.00 (reference) | **3.84** (1.18,12.47) | 0.02 | 0.76 (0.34,1.70) | 0.51 | – | |
| *Vertebral fracture* | | | | | | | |
| All | 1.00 (reference) | **2.47** (1.25,4.88) | 0.01 | **4.67** (3.59,6.08) | <0.001 | **2.75** (1.28,5.91) | 0.01 |
| Age | | | | | | | |
| 30–39 | 1.00 (reference) | 1.25 (0.15,10.57) | 0.84 | **8.60** (3.4,21.72) | <0.001 | **5.99** (0.73,48.92) | 0.09 |
| 40–49 | 1.00 (reference) | **3.00** (1.21,7.43) | 0.02 | **3.19** (2.22,4.56) | <0.001 | **2.98** (1.20,7.41) | 0.02 |
| 50–59 | 1.00 (reference) | 5.70 (0.71,45.90) | 0.10 | **10.03** (4.38,22.97) | <0.001 | 3.59 (0.44,29.31) | 0.23 |
| ≧60 | 1.00 (reference) | **6.15** (1.43,26.45) | 0.01 | **7.73** (4.45,13.43) | <0.001 | – | |
| *Other bone fracture* | | | | | | | |
| All | 1.00 (reference) | **2.83** (2.10,3.80) | <0.001 | **3.91** (3.44,4.45) | <0.001 | **4.16** (3.10,5.57) | <0.001 |
| Age | | | | | | | |
| 30–39 | 1.00 (reference) | **3.27** (1.66,6.45) | <0.001 | **6.31** (4.09,9.74) | <0.001 | **9.50** (4.38,20.63) | <0.001 |
| 40–49 | 1.00 (reference) | **2.76** (1.79,4.25) | <0.001 | **3.33** (2.83,3.92) | <0.001 | **4.18** (2.95,5.92) | <0.001 |
| 50–59 | 1.00 (reference) | **4.78** (2.30,9.95) | <0.001 | **4.51** (3.26,6.26) | <0.001 | **2.85** (1.24,6.59) | 0.01 |
| ≧60 | 1.00 (reference) | **8.49** (3.82,18.88) | <0.001 | **5.93** (4.04,8.72) | <0.001 | 1.75 (0.24,12.79) | 0.58 |

PY: Person-years; IR: Incidence rate per 1,000 person-years; HR: Hazard ratio; CI: Confidence interval.

*: Model was adjusted for age, urbanization, insurance premium, occupation, estrogen treatment, and Charlson comorbidity index.

## Discussion

This population cohort study evaluated 9,189 hysterectomized women and 33,942 matched comparisons. Both groups were primarily middle-aged women with a median age of 45. After

**Table 4. Risk of vertebral fracture in women with hysterectomy compared with the comparison group stratified by follow-up year.**

| | Comparison group | | | | Hysterectomy | | | | Crude | | Adjusted* | |
|---|---|---|---|---|---|---|---|---|---|---|---|---|
| Follow time | N | Event | PY | IR | N | Event | PY | IR | HR (95% CI) | p-value | HR (95% CI) | p-value |
| <5 | 32189 | 54 | 148499 | 0.36 | 8267 | 59 | 39044 | 1.51 | **4.37** (3.02,6.33) | <0.001 | **4.33** (2.99,6.28) | <0.001 |
| 5–9 | 32135 | 47 | 114049 | 0.41 | 8208 | 49 | 30847 | 1.59 | **4.09** (2.74,6.10) | <0.001 | **3.89** (2.60,5.82) | <0.001 |
| ≧9 | 32088 | 13 | 138354 | 0.09 | 8159 | 19 | 38228 | 0.50 | **5.71** (2.82,11.55) | <0.001 | **5.42** (2.66,11.01) | <0.001 |

PY: Person-years; IR: Incidence rate per 1,000 person-years; HR: Hazard ratio; CI: Confidence interval.

*: Model was adjusted for age, urbanization, insurance premium occupation, estrogen treatment, and Charlson comorbidity index.

**Table 5. Risk of other bone fractures in women with hysterectomy compared with the comparison group stratified by follow-up year.**

| | Comparison group | | | | Hysterectomy | | | | Crude | | Adjusted* | |
|---|---|---|---|---|---|---|---|---|---|---|---|---|
| Follow time | N | Event | PY | IR | N | Event | PY | IR | HR (95% CI) | p-value | HR (95% CI) | p-value |
| <5 | 32602 | 299 | 148499 | 2.01 | 8643 | 283 | 39044 | 7.25 | **3.69** (3.13,4.34) | <0.001 | **3.61** (3.06,4.25) | <0.001 |
| 5–9 | 32303 | 159 | 114049 | 1.39 | 8360 | 154 | 30847 | 4.99 | **3.76** (3.01,4.7) | <0.001 | **3.68** (2.95,4.60) | <0.001 |
| ≧9 | 32144 | 69 | 138354 | 0.50 | 8206 | 66 | 38228 | 1.73 | **3.73** (2.66,5.22) | <0.001 | **3.69** (2.63,5.17) | <0.001 |

PY: Person-years; IR: Incidence rate per 1,000 person-years; HR: Hazard ratio; CI: Confidence interval.

*: Model was adjusted for age, urbanization, insurance premium, occupation, estrogen treatment, and Charlson comorbidity index.

a median of 6.66 years of follow-up, women with hysterectomy had an overall 2.26-fold higher risk of developing osteoporosis or fracture. Furthermore, the hysterectomized women had a 4.92-fold higher risk of vertebral fracture compared with the comparison group.

Anti-Mullerian hormone (AMH) was used to quantify the ovarian reserve [21]. The normal value is between 2–4 ng/ml [22]. Several longitudinal studies about ovary-sparing hysterectomy with ovarian reserve have shown that premenopausal hysterectomy can cause earlier ovarian failure and decrease AMH levels one year after the procedure [23,24]. There are multiple theories of why hysterectomy with ovarian reserve leads to ovarian failure, including decreased blood flow to the ovaries after utero-ovarian ligament ligation, paracrine or endocrine effects from the uterus to the ovary, or an increase in uterus inhibition of pituitary follicle-stimulating hormone [12]. The previous study explored whether laparoscopic hysterectomy could affect ovarian reserve compared to non-laparoscopic hysterectomy. They showed both kinds of hysterectomy could decrease AMH [25]. Moreover, in the laparoscopy group, the cause of decreasing AMH may be due to electrocauterization. A randomized trial of hysterectomy with or without salpingectomy also showed decreased AMH (from 1.44 to 1.13 ng/ml) after both kinds of surgeries [26]. However, we did not have data of the AMH of each woman in our database. In our study, we found hysterectomy itself could be associated with an increased risk of osteoporosis and fracture might related to a decreased AMH.

Premenopausal oophorectomy can cause immediate surgical-related menopause. Ovarian dysfunction contributes to bone mineral density decline and increases the risk of osteoporosis and fracture [27,28]. The previous study showed postmenopausal women received BSO, the risk of fracture increased than the expected fracture rate (standardized incidence ratio [SIR], 1.54; 95% CI, 1.29–1.82) [29]. They concluded postmenopausal androgen produced by ovary may associated with a decreased incidence of fracture. However, a prospective cohort study examined the association between hysterectomy plus BSO and hip fracture risk, they found BSO were not associated with an increased risk of hip fracture (HR = 0.83 [95% CI = 0.63–1.10]) [30]. The same with the above study, our study found women with hysterectomy plus bilateral oophorectomy associated with an increased risk of osteoporosis or bone fracture.

Hormone therapy (HT) may decrease the risk of osteoporosis and bone fracture [31]. One systematic review including 28 studies had been shown the overall relative risk of HT was 0.74 (95% CI 0.69–0.80) for total fractures, 0.72 (95% CI 0.53–0.98) for hip fractures, and 0.63 (95% CI 0.44–0.91) for vertebral fractures [31]. However, the other study showed estrogen therapy was not associated with a reduction in overall fracture risk (hazard ratio [HR], 0.90; 95% CI, 0.64–1.28) and osteoporotic fractures (HR, 0.80; 95% CI, 0.52–1.23) [29]. Another study also showed the standard dose of HT was not adequate for bone mineral density in premature ovarian failure women [32]. In this study, we found the association between estrogen therapy and the risk of osteoporosis and bone fracture.

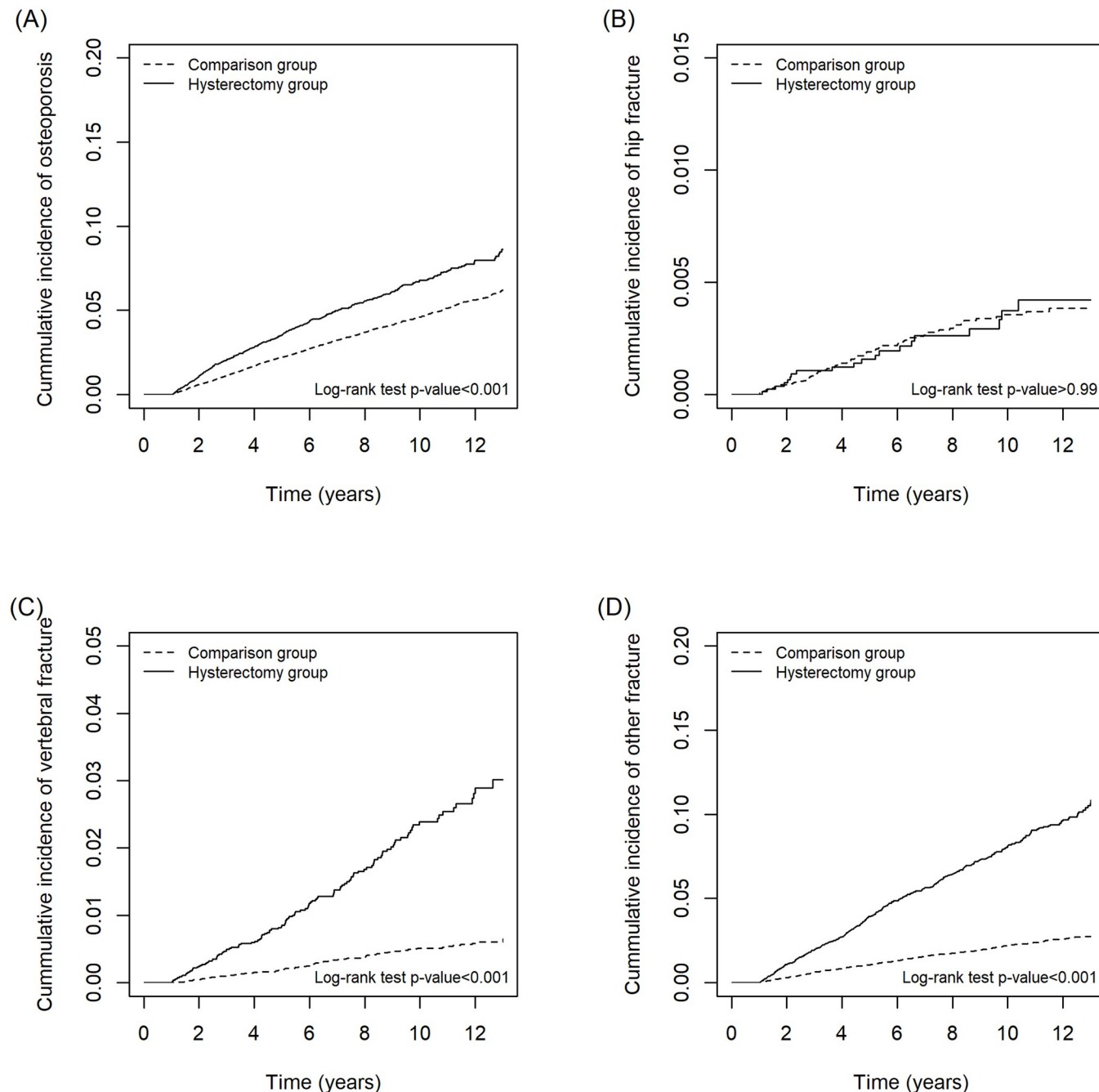

**Fig 2.** Kaplan–Meier curves showing the cumulative incidence of (A) osteoporosis, (B) hip fracture, (C) vertebral fracture, and (D) other fracture in women receiving hysterectomy (dashed line) compared with the age- and comorbidity-matched comparison group (solid line).

A national cohort study from South Korea revealed that osteoporosis had an aHR of 1.45 in the hysterectomy group, which was similar to our study [12]. However, the risk of the major complication was in osteoporosis and fracture was not investigated. In our study, we found a nearly 4.5-fold increased risk of vertebral fracture in the hysterectomy group. Furthermore, vertebral fracture had the highest risk during more than nine years of long-term follow-up.

**Table 6. The incidence rate ratio of osteoporosis or fracture interacts with estrogen treatment.**

| Hysterectomy | Estrogen treatment | N | Event | PY | IR | Crude | p-value | Adjusted* | p-value |
|---|---|---|---|---|---|---|---|---|---|
| | | | | | | IRR (95% CI) | | | |
| | | | | | *Osteoporosis or bone fracture* | | | | |
| No | No | 33833 | 1852 | 251443 | 7.37 | 1.00 (reference) | | 1.00 (reference) | |
| No | Yes | 59 | 15 | 450 | 33.31 | **4.65** (2.80,7.73) | <0.001 | **3.49** (2.10,5.81) | <0.001 |
| Yes | No | 9151 | 1043 | 64012 | 16.29 | **2.09** (1.93,2.25) | <0.001 | **2.05** (1.90,2.21) | <0.001 |
| Yes | Yes | 38 | 6 | 208 | 28.85 | **2.89** (1.30,6.44) | <0.001 | **2.74** (1.23,6.12) | 0.01 |
| | | | | | *Osteoporosis* | | | | |
| No | No | 33180 | 1149 | 247968 | 4.63 | 1.00 (reference) | | 1.00 (reference) | |
| No | Yes | 49 | 5 | 398 | 12.57 | **2.95** (1.22,7.09) | 0.02 | 2.18 (0.90,5.25) | 0.08 |
| Yes | No | 8527 | 419 | 60769 | 6.90 | **1.42** (1.27,1.59) | <0.001 | **1.41** (1.26,1.58) | <0.001 |
| Yes | Yes | 34 | 2 | 193 | 10.36 | 1.70 (0.42,6.80) | 0.45 | 1.55 (0.39,6.20) | 0.54 |
| | | | | | *Hip fracture* | | | | |
| No | No | 32111 | 80 | 242384 | 0.33 | 1.00 (reference) | | 1.00 (reference) | |
| No | Yes | 46 | 2 | 374 | 5.35 | **17.45** (4.29,70.99) | <0.001 | **9.59** (2.33,39.58) | 0.002 |
| Yes | No | 8128 | 20 | 58914 | 0.34 | 0.99 (0.61,1.61) | 0.96 | 1.01 (0.62,1.64) | 0.98 |
| Yes | Yes | 32 | 0 | 187 | 0.00 | – | | – | |
| | | | | | *Vertebral fracture* | | | | |
| No | No | 32139 | 108 | 242607 | 0.45 | 1.00 (reference) | | 1.00 (reference) | |
| No | Yes | 50 | 6 | 402 | 14.94 | **35.71** (15.7,81.25) | <0.001 | **26.33** (11.44,60.60) | <0.001 |
| Yes | No | 8235 | 127 | 59515 | 2.13 | **4.59** (3.55,5.93) | <0.001 | **4.45** (3.44,5.76) | <0.001 |
| Yes | Yes | 32 | 0 | 187 | 0.00 | – | | – | |
| | | | | | *Other bone fracture* | | | | |
| No | No | 32556 | 525 | 244617 | 2.15 | 1.00 (reference) | | 1.00 (reference) | |
| No | Yes | 46 | 2 | 369 | 5.42 | 2.70 (0.67,10.81) | 0.16 | 2.59 (0.65,10.41) | 0.18 |
| Yes | No | 8607 | 499 | 61372 | 8.13 | **3.60** (3.18,4.06) | <0.001 | **3.53** (3.12,3.99) | <0.001 |
| Yes | Yes | 36 | 4 | 202 | 19.81 | **6.89** (2.58,18.43) | <0.001 | **6.73** (2.51,18.05) | <0.001 |

PY: Person-years; IR: Incidence rate per 1,000 person-years; HR: Hazard ratio; CI: Confidence interval.

*: Model was adjusted for age, urbanization, insurance premium, occupation, estrogen treatment, and Charlson comorbidity index.

Bone density decline is the main reason for osteoporotic fracture, and it independently increases the incidence of fracture [33]. Vertebral and femoral fractures are the two leading locations of osteoporotic fractures. According to a worldwide study, vertebral fractures comprise 16% of total osteoporotic fractures [34]. Furthermore, women with a vertebral fracture experience a 3.7 times higher mortality rate during the first year after vertebral fracture compared with those who did not have a vertebral fracture.

Controversially, a population-based cohort study in 2008 reported that hysterectomy elevated overall fracture risk, but the only statistically significant increases were found for fractures in the hands and feet. No significant increase in fractures was found in traditional osteoporotic fracture sites such as the hip, spine, or distal forearm [11]. A possible reason for these different results is that this study included cancer and pre-cancerous conditions in their operation indication, which might affect the fracture risk.

## Strengths and limitations

Our study is population-based assessment research, and this study design can minimize selection bias. Additionally, the study data were adjusted through conventional medical histories, related comorbidities, and comorbidity severity. However, our study has some potential

limitations. First, medications or supplements that may be related to osteoporosis or fracture were not considered. There are several different types of supplements that claim to prevent osteoporosis, such as fish oil, Vitamin D, glucosamine, etc. Second, lab data including bone mineral density, calcium, magnesium, and phosphate levels were not collected in the TNHI database. Third, the diagnostic code for fractures could not differentiate between osteoporotic fractures or fractures caused by other reasons. Instead, we considered the two leading sites of osteoporotic fractures, hip and vertebral, in our study subjects. Fourth, the proportion of CCI at 1 or 2 was higher in the hysterectomy group than in the comparison group, which may be a potential confounder. Body mass index, alcohol consumption, history of endocrine disease, physical inactivity, medication, eating habits, family history of osteoporosis, and smoking history are also important factors in the risk of osteoporosis. However, they were not recorded in the database. Taking oral contraceptives may also contribute to the risk of osteoporosis. However, the database also did not record the prescription of oral contraceptives due to self-paid.

## Conclusion

Hysterectomy might be associated with the slightly increased risk of osteoporosis and vertebral fracture in middle-aged women. Based on the study results, women who undergo hysterectomy should be screened more readily or counseled regarding this risk of osteoporosis or fracture.

## Author Contributions

**Conceptualization:** Dah-Ching Ding.

**Data curation:** Ying-Ting Yeh, Pei-Chen Li, Kun-Chi Wu, Yu-Cih Yang, Weishan Chen, Hei-Tung Yip, Dah-Ching Ding.

**Formal analysis:** Yu-Cih Yang, Weishan Chen, Hei-Tung Yip, Dah-Ching Ding.

**Funding acquisition:** Dah-Ching Ding.

**Investigation:** Weishan Chen, Dah-Ching Ding.

**Methodology:** Weishan Chen, Dah-Ching Ding.

**Software:** Yu-Cih Yang, Weishan Chen.

**Supervision:** Jen-Hung Wang, Shinn-Zong Lin, Dah-Ching Ding.

**Validation:** Jen-Hung Wang, Shinn-Zong Lin, Dah-Ching Ding.

**Visualization:** Shinn-Zong Lin, Dah-Ching Ding.

**Writing – original draft:** Ying-Ting Yeh, Pei-Chen Li, Kun-Chi Wu, Yu-Cih Yang, Weishan Chen, Hei-Tung Yip, Dah-Ching Ding.

**Writing – review & editing:** Dah-Ching Ding.

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
