## [Decision Letter · Decision Letter 0]

16 Oct 2020

PONE-D-20-30628

Hysterectomies are associated with an increased risk of osteoporosis and bone fracture: a population-based cohort study

PLOS ONE

Dear Dr. Ding,

Thank you for submitting your manuscript to PLOS ONE. After careful consideration, we feel that it has merit but does not fully meet PLOS ONE’s publication criteria as it currently stands. Therefore, we invite you to submit a revised version of the manuscript that addresses the points raised during the review process.

Both reviewers had some concerns, especially regarding the data analysis and definition. I hope the authors can effectively respond to these comments in their revision.

We look forward to receiving your revised manuscript.

Kind regards,

Yu Ru Kou, PhD

Academic Editor

PLOS ONE

Journal Requirements:

2.Thank you for stating the following in the Acknowledgments Section of your manuscript:

[This study is supported in part by the Taiwan Ministry of Health and Welfare Clinical Trial

Center (MOHW108-TDU-B-212-133004), China Medical University Hospital, Academia

Sinica Stroke Biosignature Project (BM10701010021), MOST Clinical Trial Consortium for

Stroke (MOST 107-2321-B-039 -004-), Tseng-Lien Lin Foundation, Taichung, Taiwan, and

Katsuzo and Kiyo Aoshima Memorial Funds, Japan. The funders had no role in study design,

data collection and analysis, decision to publish, or preparation of the manuscript.]

 [ The funders had no role in study design, data collection and analysis, decision to publish, or preparation of the manuscript.]

Reviewers' comments:

Reviewer's Responses to Questions

**Comments to the Author**

1. Is the manuscript technically sound, and do the data support the conclusions?

Reviewer #1: Yes

Reviewer #2: Partly

2. Has the statistical analysis been performed appropriately and rigorously? 

Reviewer #1: Yes

Reviewer #2: Yes

3. Have the authors made all data underlying the findings in their manuscript fully available?

Reviewer #1: Yes

Reviewer #2: Yes

4. Is the manuscript presented in an intelligible fashion and written in standard English?

Reviewer #1: Yes

Reviewer #2: Yes

5. Review Comments to the Author

Reviewer #1: Major revision

1. In the flowchart, 4-fold size matched of women without oophorectomy used index date; however, 4-fold size matched of women with oophorectomy used index year. Why the criterion was different?

2. In the flowchart, there were 879 women with hysterectomy and oophorectomy. So the 4-fold size matched of women in comparison group with oophorectomy was 3,516 (879×4). But the number of your research was 1,755, please explain the reason.

3. Osteoporosis, one of the outcomes, was defined by ICD-9-CM 733. However, cyst of bone (ICD-9-CM 733.2), hyperostosis of skull (ICD-9-CM 733.3), aseptic necrosis of bone (ICD-9-CM 733.4), osteitis condensans (ICD-9-CM 733.5), Tietze's disease (ICD-9-CM 733.6), algoneurodystrophy (ICD-9-CM 733.7), malunion and nonunion of fracture (ICD-9-CM 733.8), and other and unspecified disorders of bone and cartilage (ICD-9-CM 733.9) were not direct associated with osteoporosis or fracture. Besides, the mechanism of osteoporosis (ICD-9-CM 733.0) and pathologic fracture (ICD-9-CM 733.1) was different. Therefore, the outcomes in this study should revise to osteoporosis (ICD-9-CM 733.0) or fracture (pathologic fracture, ICD-9-CM 733.1; hip fracture, ICD-CM 820-821; vertebral fracture. ICD-9-CM 805-806; and other fracture, ICD-9-CM 800-829).

Minor revision

1. Please provide the information about years from index date through outcomes in study and comparison cohort.

2. Injury was the causes of fractures, so the study should adjust for motor vehicle injury (ICD-9-CM E810-E819) and falls (ICD-9-CM E880-E888) in the regression model.

3. Hysterectomy (ICD-9-CM codes 97020K…...) should revise to hysterectomy (NHI claim codes 97020K…...).

4. Unilateral or bilateral oophorectomy (ICD-9-CM code 80802C……) should revise to unilateral or bilateral oophorectomy (NHI claim codes 80802C…...).

5. In the NHIRD, there was no income data. It should revise monthly income to insurance premium.

6. In the NHIRD, there were only insured classification and group insurance applicant information. How to definition of white collar and blue collar.

7. In the table, please unify the decimal places of % and P value.

Reviewer #2: Comment 1: This study aimed to investigate-- Hysterectomies are associated with an increased risk of osteoporosis and bone fracture: a population-based cohort study

. The authors provided a sound background for their investigation then used sound statistical methods in this investigation, such as survival curves and Cox regression model to draw appropriate conclusions.

Comment 2: The authors used statistical methods that are sound. Chi-square tests and Wilcoxon rank-sum tests to examine the differences between categorical and continuous baseline characteristics, survival curves and Cox regression model were used to perform the data analyses in this study.

Comment 3: Please tone down the conclusion in your abstract so that it does not overstate your results, which cannot at this stage be generalized beyond the study population.

Comment 4: Abstract, Aim: This study investigated the risk of osteoporosis or bone fractures (vertebral, hip and others) in hysterectomized women in Taiwan. The word " vertebral " is an adjective word.

Comment 5: Materials and methods,Outcomes: In Taiwan, the diagnosis of osteoporosis was made by a bone density scan (DEXA) exam. The sentence " bone density scan (DEXA) exam "should switch to " Dual-energy X-ray absorptiometry (DXA) exam ". The word " vertebral " is an adjective word.

Comment 6: The authors mentioned hysterectomies were associated with slightly increased risks of developing osteoporosis or a vertebral fracture. Please explain other confounding factors, such as BMI, alcohol consumption, history of endocrine disease, physical inactivity, medication, eating habits, family history of osteoporosis and smoking history are also important factors in the risk of osteoporosis.

6. PLOS authors have the option to publish the peer review history of their article (what does this mean?). If published, this will include your full peer review and any attached files.

Reviewer #1: No

Reviewer #2: No

---

## [Author Response · Author response to Decision Letter 0]

13 Nov 2020

Re: PONE-D-20-30628

Hysterectomies are associated with an increased risk of osteoporosis and bone fracture: a population-based cohort study

Reviewer #1: Major revision

1. In the flowchart, 4-fold size matched of women without oophorectomy used index date; however, 4-fold size matched of women with oophorectomy used index year. Why the criterion was different?

Response: Thanks for asking. We changed the “index date” to “index year” in Figure 1 and the Method section (p. 6, line 4 and 6).

2. In the flowchart, there were 879 women with hysterectomy and oophorectomy. So the 4-fold size matched of women in comparison group with oophorectomy was 3,516 (879×4). But the number of your research was 1,755, please explain the reason.

Response: Thanks for asking. The number of women in the comparison group with oophorectomy was 1684 was because of the insufficient number of women with oophorectomy. Therefore, there was not completely 1:4 matching.

3. Osteoporosis, one of the outcomes, was defined by ICD-9-CM 733. However, cyst of bone (ICD-9-CM 733.2), hyperostosis of skull (ICD-9-CM 733.3), aseptic necrosis of bone (ICD-9-CM 733.4), osteitis condensans (ICD-9-CM 733.5), Tietze's disease (ICD-9-CM 733.6), algoneurodystrophy (ICD-9-CM 733.7), malunion and nonunion of fracture (ICD-9-CM 733.8), and other and unspecified disorders of bone and cartilage (ICD-9-CM 733.9) were not direct associated with osteoporosis or fracture. Besides, the mechanism of osteoporosis (ICD-9-CM 733.0) and pathologic fracture (ICD-9-CM 733.1) was different. Therefore, the outcomes in this study should revise to osteoporosis (ICD-9-CM 733.0) or fracture (pathologic fracture, ICD-9-CM 733.1; hip fracture, ICD-CM 820-821; vertebral fracture. ICD-9-CM 805-806; and other fracture, ICD-9-CM 800-829).

Response: Thanks for the suggestion. We revised as the reviewer suggested (p. 6, line 18-20).

Minor revision

1. Please provide the information about years from index date through outcomes in study and comparison cohort.

Response: Thanks for the suggestion. In the hysterectomy group (n=1049), the years (mean [SD]) from the index date through outcome was 4.97 [3.01] years, while in the comparison group (n=1867), there were 5.09 [3.02] years (p. 8, line 16-18). 

2. Injury was the causes of fractures, so the study should adjust for motor vehicle injury (ICD-9-CM E810-E819) and falls (ICD-9-CM E880-E888) in the regression model.

Response: Thanks for the suggestion. We excluded those patients with fractures due to vehicle injury and falls from the study (p. 6, line 9).

3. Hysterectomy (ICD-9-CM codes 97020K…...) should revise to hysterectomy (NHI claim codes 97020K…...).

Response: Thanks for the suggestion. We revised as the reviewer suggested (p. 6, line 2).

4. Unilateral or bilateral oophorectomy (ICD-9-CM code 80802C……) should revise to unilateral or bilateral oophorectomy (NHI claim codes 80802C…...).

Response: Thanks for the suggestion. We revised as the reviewer suggested (p. 7, line 1).

5. In the NHIRD, there was no income data. It should revise monthly income to insurance premium.

Response: Thanks for the suggestion. We revised as the reviewer suggested (Table 1).

6. In the NHIRD, there were only insured classification and group insurance applicant information. How to definition of white collar and blue collar.

Response: Occupation data can be obtained in the unit_ins_type variable in Registration files. The classification of occupations is as follows:

white collar: Civil servants and employees of central non-institutional organizations, central public servants, national colleges, private colleges, private elementary and secondary faculty and employees, government agencies, school public teachers, local public officials, non-profit enterprises or private employees, public institutions staff and workers

blue collar: Employers with certain employers, trainees from vocational training institutions, self-employed workers, self-employed workers of specialized occupations and technicians, members of professional associations, seafarers’ unions or captains’ guilds, seafarers, farmers, members of water conservancy associations, fishermen members

others: Military dependents, military students, military personnel, alternative service personnel, low-income households, veterans, veterans family members, monks, religious persons, and the person lives in social welfare institutions

7. In the table, please unify the decimal places of % and P value.

Response: Thanks for the suggestion. We revised as the reviewer suggested.

Reviewer #2: Comment 1: This study aimed to investigate-- Hysterectomies are associated with an increased risk of osteoporosis and bone fracture: a population-based cohort study

. The authors provided a sound background for their investigation then used sound statistical methods in this investigation, such as survival curves and Cox regression model to draw appropriate conclusions.

Response: Thanks for the comment.

Comment 2: The authors used statistical methods that are sound. Chi-square tests and Wilcoxon rank-sum tests to examine the differences between categorical and continuous baseline characteristics, survival curves and Cox regression model were used to perform the data analyses in this study.

Response: Thanks for the comment.

Comment 3: Please tone down the conclusion in your abstract so that it does not overstate your results, which cannot at this stage be generalized beyond the study population.

Response: Thanks for the suggestion. We have toned down the conclusion in the abstract section (p. 3, line 21).

Comment 4: Abstract, Aim: This study investigated the risk of osteoporosis or bone fractures (vertebral, hip and others) in hysterectomized women in Taiwan. The word " vertebral " is an adjective word.

Response: Thanks for the suggestion. We changed “vertebral” to “vertebrae” (p. 3, line 2).

Comment 5: Materials and methods,Outcomes: In Taiwan, the diagnosis of osteoporosis was made by a bone density scan (DEXA) exam. The sentence " bone density scan (DEXA) exam "should switch to " Dual-energy X-ray absorptiometry (DXA) exam ". The word " vertebral " is an adjective word.

Response: Thanks for the suggestion. We changed them as reviewers suggested. We changed “vertebral” to “vertebrae” (p. 6, line 21-23).

Comment 6: The authors mentioned hysterectomies were associated with slightly increased risks of developing osteoporosis or a vertebral fracture. Please explain other confounding factors, such as BMI, alcohol consumption, history of endocrine disease, physical inactivity, medication, eating habits, family history of osteoporosis and smoking history are also important factors in the risk of osteoporosis.

Response: Thanks for the suggestion. We added these risk factors in the limitation section (p. 13, line 21-22).

---

## [Decision Letter · Decision Letter 1]

16 Nov 2020

Hysterectomies are associated with an increased risk of osteoporosis and bone fracture: a population-based cohort study

PONE-D-20-30628R1

Dear Dr. Ding,

We’re pleased to inform you that your manuscript has been judged scientifically suitable for publication and will be formally accepted for publication once it meets all outstanding technical requirements.

Kind regards,

Yu Ru Kou, PhD

Academic Editor

PLOS ONE

Additional Editor Comments (optional):

Reviewers' comments:

Reviewer's Responses to Questions

**Comments to the Author**

1. If the authors have adequately addressed your comments raised in a previous round of review and you feel that this manuscript is now acceptable for publication, you may indicate that here to bypass the “Comments to the Author” section, enter your conflict of interest statement in the “Confidential to Editor” section, and submit your "Accept" recommendation.

Reviewer #1: All comments have been addressed

Reviewer #2: All comments have been addressed

2. Is the manuscript technically sound, and do the data support the conclusions?

Reviewer #1: Yes

Reviewer #2: Yes

3. Has the statistical analysis been performed appropriately and rigorously? 

Reviewer #1: Yes

Reviewer #2: Yes

4. Have the authors made all data underlying the findings in their manuscript fully available?

Reviewer #1: Yes

Reviewer #2: Yes

5. Is the manuscript presented in an intelligible fashion and written in standard English?

Reviewer #1: Yes

Reviewer #2: Yes

6. Review Comments to the Author

Reviewer #1: (No Response)

Reviewer #2: Congratulation!according to my opinion to the evaluation of this manuscript.

All of you did a good job. You have revised the article well.

7. PLOS authors have the option to publish the peer review history of their article (what does this mean?). If published, this will include your full peer review and any attached files.

Reviewer #1: No

Reviewer #2: No

---

## [Editor Report · Acceptance letter]

18 Nov 2020

PONE-D-20-30628R1 

Hysterectomies are associated with an increased risk of osteoporosis and bone fracture: a population-based cohort study 

Dear Dr. Ding:

I'm pleased to inform you that your manuscript has been deemed suitable for publication in PLOS ONE. Congratulations! Your manuscript is now with our production department. 

Kind regards, 

on behalf of

Dr. Yu Ru Kou 

Academic Editor

PLOS ONE